# DeePosit, an AI-based tool for detecting mouse urine and fecal depositions from thermal video clips of behavioral experiments

**David Peles\*, Shai Netser, Natalie Ray, Taghreed Suliman, Shlomo Wagner**

Sagol Department of Neurobiology, Faculty of Natural Sciences, University of Haifa, Haifa, Israel

## eLife Assessment

This manuscript presents an **important** machine-learning-based approach to the automated detection of urine and fecal deposits by rodents, key ethological behaviors that have traditionally been very poorly studied. The strength of evidence for the claim is **solid**, showing accuracy near 90% across several contexts. Training and testing for the specific contexts used by other experimenters, however, is probably warranted to make the model most relevant to the data that may be analyzed.

**\*For correspondence:**
davidpelz@gmail.com

**Competing interest:** The authors declare that no competing interests exist.

**Abstract** In many mammals, including rodents, social interactions are often accompanied by active urination (micturition), which is considered a mechanism for spatial scent marking. Urine and fecal deposits contain a variety of chemosensory signals that convey information about the individual's identity, genetic strain, social rank, and physiological or hormonal state. Furthermore, scent marking has been shown to be influenced by the social context and by the individual's internal state and experience. Therefore, analyzing scent-marking behavior during social interactions can provide valuable insight into the structure of mammalian social interactions in health and disease. However, conducting such analyses has been hindered by several technical challenges. For example, the widely used void spot assay lacks temporal resolution and is prone to artifacts, such as urine smearing. To solve these issues, recent studies employed thermal imaging for the spatio-temporal analysis of urination activity. However, this method involved manual analysis, which is time-consuming and susceptible to observer bias. Moreover, defecation activity was hardly analyzed by previous studies. In the present study, we integrate thermal imaging with an open-source algorithm based on a transformer-based video classifier for automatic detection and classification of urine and fecal deposits made by male and female mice during various social behavior assays. Our results reveal distinct dynamics of urination and defecation in a test-, strain-, and sex-dependent manner, indicating two separate processes of scent marking in mice. We validate this algorithm, termed by us DeePosit, and show that its accuracy is comparable to that of a human annotator and that it is efficient in various setups and conditions. Thus, the method and tools introduced here enable efficient and unbiased automatic spatio-temporal analysis of scent-marking behavior in the context of behavioral experiments in small rodents.

## Introduction

In many mammalian species, including rodents, social interactions are accompanied or followed by events of active urination, also known as micturition or voiding activity (*Arakawa et al., 2008*). Multiple

**eLife digest** Scientists conduct behavioral experiments on animals to study brain mechanisms that govern social behavior and how these may be affected by various conditions. For example, in rodents, urination and defecation are important social activities used for communication and territory marking, and they are influenced by the emotional state of an individual.

In the past, these activities were analyzed at the end of an experiment by shining ultraviolet light on a filter paper placed on the floor of the cages. However, this method does not provide information on when urination or defecation occurred. Also, in many cases, urine drops are smeared on the filter paper due to the animal's movement during the experiment, which reduces the accuracy of this method. To bridge this gap, Peles et al. developed a computer-vision algorithm – named DeePosit – to automatically track mice's urination and defecation activities during social behavior experiments recorded with a thermal camera.

To examine the efficiency of the tool, the researchers analyzed the urination and defecation activities of mice during several social behavior tests. They then tested whether these activities changed over time and if there were differences between male and female mice, or between different strains of laboratory mice.

The analysis revealed that the tool could identify the time and location of each urination and defecation event with an accuracy similar to that of a human observer. Using this tool, Peles et al. demonstrated that urination and defecation activities changed during a social encounter, for example, urination became more frequent. They observed that males urinated more often than females, which may be attributed to differences in their territorial behavior. It also revealed differences between laboratory strains.

Peles et al. are confident that this rapid, unbiased and cost-effective tool can improve the analysis of social behavior in animals, particularly rodents. This will be especially relevant for researchers investigating the effect of treatments in mouse models of various disorders. The tool can also be trained and adapted to different behavioral and experimental contexts. It may allow a comparison of an additional important aspect of social behavior in treated and non-treated animals, and in health and disease.

studies have demonstrated that urine and fecal deposits comprise many chemosensory social signals that carry information about the individual, such as its species, sex, social rank, and identity, as well as its reproductive and health conditions (*Bigiani et al., 2005*). These chemosensory signals include various metabolites, as well as many proteins, such as major urinary proteins (*Brennan, 2004*). Thus, by depositing urine spots and feces in its environment, the individual also deposits social information, which may later be perceived by other individuals and modify their future social interactions with this individual (*Hurst and Beynon, 2004*). In other words, the use of urine and fecal deposits allows individuals to advertise their availability to possible mates and communicate with other conspecifics. Moreover, in territorial species, urination is used to mark the territory of the individual, thus functioning as a spatio-social scent-marking activity (*Brennan and Kendrick, 2006*). In rodents, urination was shown to be strongly influenced by the individual's internal state, social rank, social context, and previous social experience (*Desjardins et al., 1973*; *Hyun et al., 2021*). Therefore, monitoring urination activity can provide valuable information on the individual's social behavior and internal state. Specifically, deficits in urine depositing may reflect atypical social behavior in rodent models of various diseases (see *Wöhr et al., 2011* for example), hence may be used for testing potential treatments in such models.

Urination during a given task is traditionally analyzed via the void spot assay, which uses filter paper placed on the arena floor to analyze, after the end of the experiment, the spatial distribution of urine spots (*Wolff and Powell, 1984*; *Higuchi and Arakawa, 2022*). However, this analysis usually lacks the temporal dimension, is distorted by urine smearing across the arena floor caused by the individual's movement (see Figure 2d, e), and is limited in detecting overlapping urine spots. Another caveat is that the filter paper may be torn down by the mouse during the behavioral experiment. Recently, *Dalghi et al., 2023* used a filter paper on the arena floor, UV light, several cameras, and a manual video annotation to analyze urination events. Several other studies (*Verstegen et al., 2020*; *Miller*

*et al., 2023a*) used thermal imaging via infrared (IR) camera for such analysis, as urine deposits are emitted while being in body temperature, hence can be seen in the thermal image. However, fecal deposits are also emitted in body temperature, making it difficult to distinguish between feces and small urine spots by thermal imaging alone. Moreover, these studies relied on manual analysis of thermal video clips, which made the analysis process time-consuming and subjected to observer bias. To cope with these limitations, we have developed an open-source computer vision-based software to automatically detect and classify deposited urine and feces from thermal video clips. Our detection and classification algorithm is based on a combination of a heuristic algorithm used for the preliminary detection of bright (warm) blobs in the thermal video clip and a trainable video classifier used to classify the preliminary detections as either urine, feces, or background (BG, i.e., not urine or feces). We demonstrate the efficiency of this tool by analyzing the temporal dynamics of urination and defecation activities in male and female CD1 (ICR) mice while performing three social behavior tests, and further validate the algorithm by testing it with male C57BL/6J mice. We found that urination and defecation activities show distinct dynamics across the various tests in a sex-, strain-, and test-dependent manner.

## Results

### Social discrimination

Each CD1 subject animal performed three different social discrimination tests, as previously described by *Mohapatra et al., 2024*, on three consecutive days in the order described below. Each test consisted of a 15-min habituation stage, during which the subject mouse got used to an experimental arena containing empty chambers at randomly chosen opposite corners. After habituation, the empty chambers were replaced with similar chambers containing stimuli for a 5-min trial stage (*Figure 1a*). In the **Social Preference** (SP) test, a novel (i.e., unfamiliar to the subject mouse) sex-matched stimulus mouse was placed in one chamber, while an object stimulus (a Lego toy) was placed in the opposite chamber. In the **Sex Preference** (SxP) test, a novel female mouse was placed in one chamber while a novel male was placed in the opposite chamber. In the stress version of the **Emotional State Preference** (ESPs) test, a novel stressed (restrained for 15 min before the test) mouse was introduced into one chamber while a naïve mouse was placed in the opposite chamber. We first analyzed the time spent by the subject mouse on investigating each stimulus during the three tests (*Figure 1*), using the video clips recorded via the visible light (VIS) camera. Both male and female mice showed the behavior expected from CD1 mice, as previously described by us (*Kopachev et al., 2022*). Males showed a significantly higher investigation time toward the social stimulus, as compared to the object in the SP test, toward the opposite sex, as compared to the same sex stimulus mouse in the SxP test, and toward the stressed mouse, as compared to the naïve mouse in the ESPs test. Females showed similar behavior, except for the SxP test, where they exhibited no preference for any of the two stimuli. In accordance with our previous study (*Netser et al., 2017*), in all cases, the preference toward a given stimulus was reflected only by long (>6s), but not by short (≤ 6s) investigation bouts (*Figure 1*). Thus, in terms of social behavior, the subject mice behaved as expected.

### Urine and feces detection

The experimental setup used for the detection of urine and fecal deposits, comprising VIS and IR cameras, as well as a black body, is schematically shown in *Figure 2a*. Unlike the VIS camera (*Figure 2b*), the IR camera captures the warm urine and feces drops soon after they were deposited (*Figure 2c*). This allowed us to overcome several caveats of the void spot assay. For example, we could tolerate smeared urine spots (*Figure 2d, e*) and identify the exact time of each urine or fecal deposition event. Using the thermal video clips, we designed a detection algorithm (termed DeePosit) consisting of two main parts: (1) A preliminary heuristic detection algorithm detects warm blobs (*Figure 2f*). (2) These blobs are then fed into a machine learning-based classifier (*Figure 2g*), which classifies them as urine, feces, or background (i.e., without detection) (see methods, *Videos 1 and 2*, and *Figure 2—video 1*).

For the generation of training and testing datasets, a human annotator manually tagged urination and defecation events in 157 thermal video clips (about 20 min each), of which 97 were used for training and 60 for testing. The precision, recall, and F1 score of the DeePosit algorithm for the test video clips are 0.90, 0.86, 0.88 for urine deposits and 0.91, 0.89, 0.90 for feces, respectively. The mean F1 score: $(F1_{Urine} + F1_{Feces})/2$ is 0.89 and the confusion matrix is shown in *Figure 2h*. Notably,

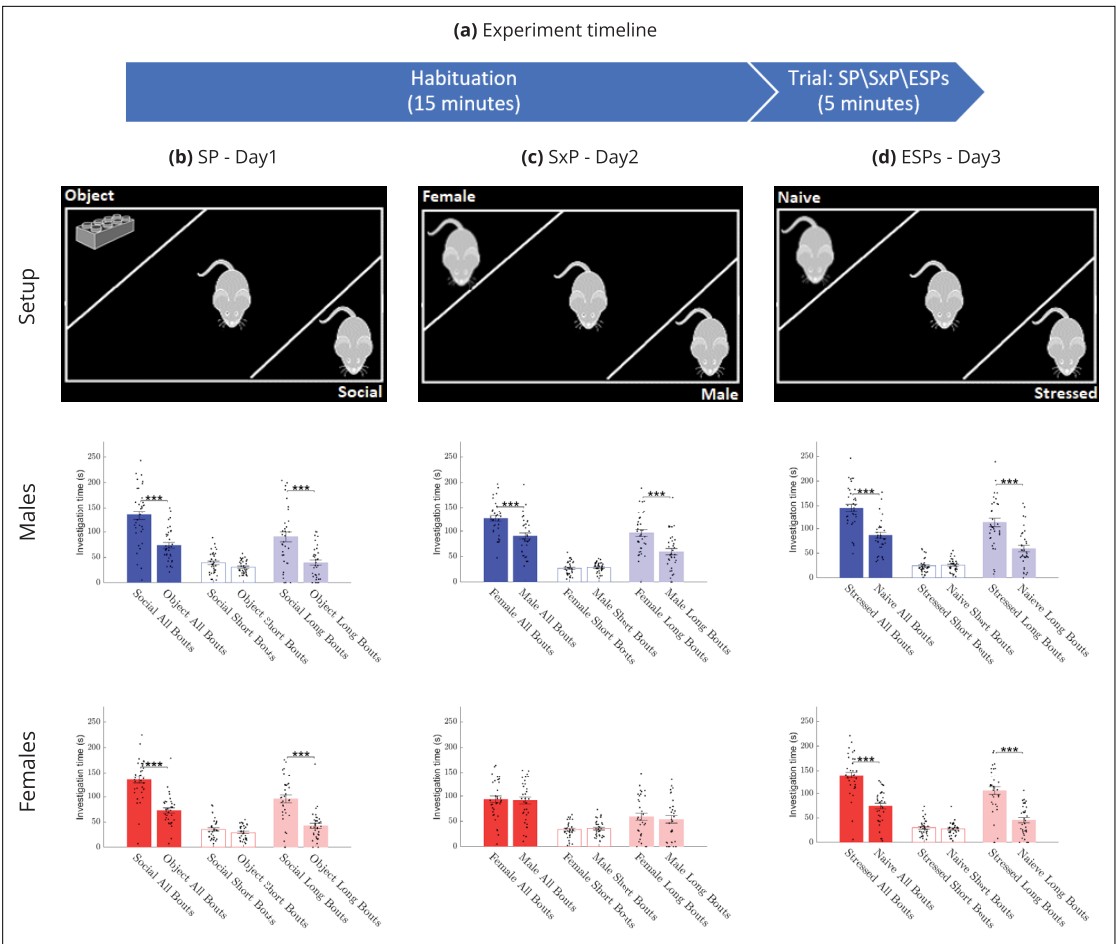

**Figure 1.** Investigation time across sexes and tests in CD1 mice. Each of the tests (SP, SxP, and ESPs) is comprised of a 15-min habituation stage with empty chambers, followed by a 5-min trial stage in which the stimuli are present in the chambers (**a**). The setup row shows schematic representations of the arena for the (**b**) SP, (**c**) SxP, and (**d**) ESPs tests, while the males and females rows show the mean (± SEM) time dedicated by male (*n* = 36, blue bars) and female (*n* = 35, red bars) mice to investigate each stimulus during the various tests. The two leftmost bars in each panel show the total investigation time, while the two middle bars show the time spent on short (≤ 6 s) investigation bouts, and the two rightmost bars show the time spent on long (>6 s) investigation bouts. A two-sided Wilcoxon rank sum test was used for statistical significance. ***p < 0.001.

for large urine deposits, the classification precision is higher (0.98), in comparison to small urine precision (0.85), most probably because large urine drops are more distinguishable from fecal deposits, which are always small *Figure 2—figure supplement 1*. See *Figure 2—video 1* and *Figure 3—figure supplement 3*, for examples of correct detections, as well as mistakes made by the detection algorithm in the test videos, which are further discussed in the Discussion section.

## Detection stability and consistency

We tested the algorithm's accuracy across various stages of the experiment (*Figure 3a, b*), the various experiments (*Figure 3c*), the two sexes (*Figure 3d*), and three equal spatial divisions of the arena (*Figure 3e*). We found that the accuracy was stable in all cases, with no significant difference between them. These results suggest that the accuracy level of the algorithm is uniform across all these instances, hence the algorithm's mistakes should not create a bias that may affect the experimental results.

We further compared the accuracy level of DeePosit with that of a second human annotator, using the first human annotator as a ground truth to both. For that, we used a subset of 25 video clips from the entire test set. The accuracy achieved by DeePosit with this dataset was comparable to that of the second human annotator (mean F1 score of 0.84 and 0.86, respectively, *Figure 3f, g*). These results

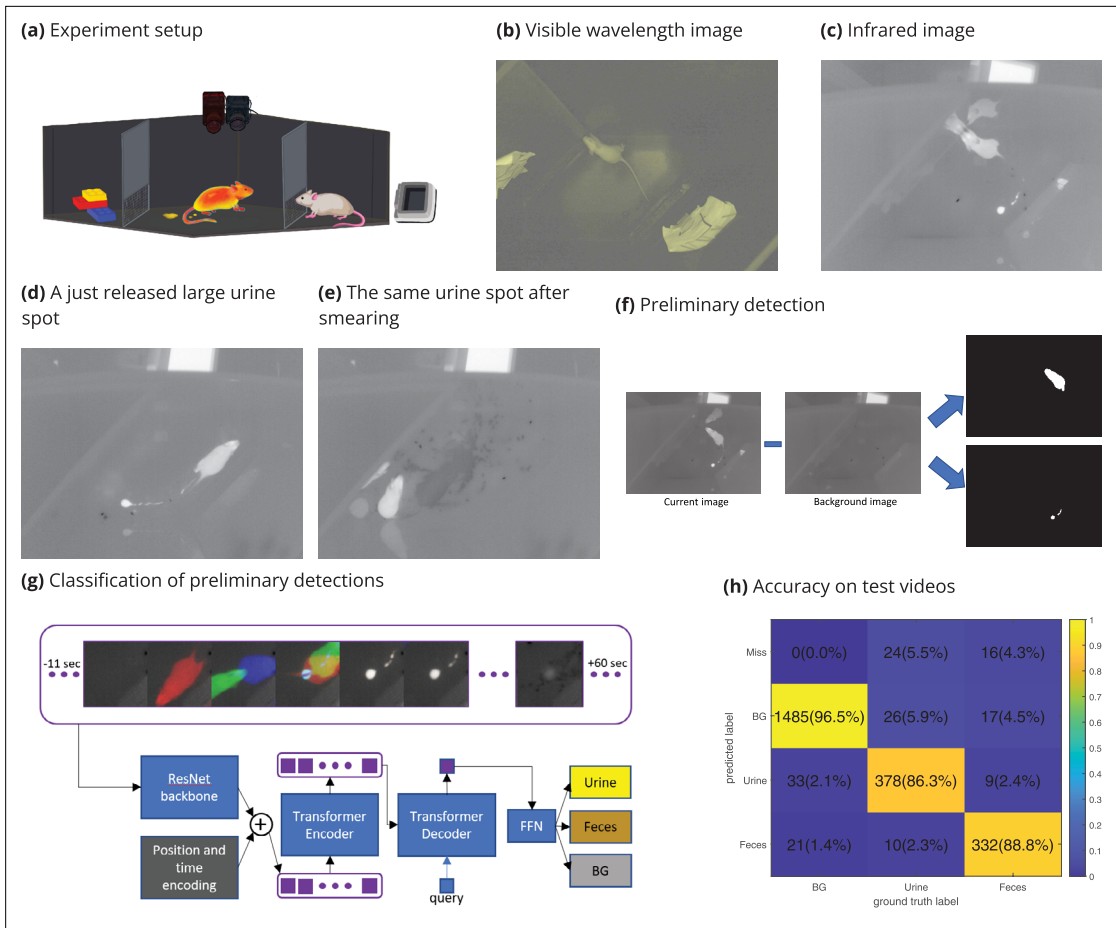

**Figure 2.** The experimental setup and analysis method. The experimental setup (**a**) includes a visible light (VIS) camera, an infrared (IR) camera, and a blackbody set to 37°C. VIS (**b**) and IR (**c**) images that were captured at the same moment, a short time after a urine deposition, exemplify that, as the urine is still warm, it appears as highly contrasted blob in the IR image but not in the VIS one. Large urine spots, such as the one shown in (**d**), may be smeared across the arena's floor (**e**), which is one limitation of the use of filter paper for quantifying urination at the end of the experiment. The preliminary detection algorithm is based on subtracting a background image from each frame in the video (**f**), which allows the detection of hot blobs reflecting the animal itself and urine and feces deposits. The detected blobs are then classified using a transformer-based artificial neural network (**g**), which gets as its input a time series of patches cropped around the detection and provides its classification as an output. Each three patches in that time series are merged into a single RGB image (see methods). In the confusion matrix presenting the accuracy of the full pipeline for test videos (**h**) in CD1 mice, the 'Miss' row counts the events that were not detected by the preliminary hot blobs detection and, hence, were not fed to the classifier. The BG (background) column counts the number of automatic detections for which no matching manually tagged event exists in the relevant space and time window. Test videos include videos from 60 experiments. See Methods for more details. The precision, recall, and F1 score for urine detection are 0.90, 0.86, 0.88 accordingly, and 0.91, 0.89, 0.90 for feces detection. The mean F1 score: $(F1_{Urine} + F1_{Feces})/2$ is 0.89.

The online version of this article includes the following video and figure supplement(s) for figure 2:

**Figure 2—video 1.** Video for the events in the confusion matrix.

https://elifesciences.org/articles/100739/figures#fig2video1

**Figure supplement 1.** Accuracy for small and large detections in CD1 mice.

demonstrate the partial accuracy of urination and defecation annotation by human observers and show that DeePosit is comparable to a trained observer in tagging urine and fecal depositions.

We also compared the accuracy of DeePosit with the accuracy achieved by a classic object detection algorithm (YOLOv8) (*Jocher et al., 2023*). For that, we annotated 39 training videos of CD1 mice with bounding boxes to match the YOLOv8 framework. For fairness, we compared YOLOv8 results with DeePosit algorithm that was trained on the same set of video clips. DeePosit was significantly better (mean F1 = 0.81) than YOLOv8, regardless whether we used a single image (YOLOv8 Gray, F1 = 0.58), or a sequence of three images (0, 10, and 30 s after each frame, YOLOv8 RGB, F1 = 0.68) as in input (see *Figure 3h–j*). The fact that using a sequence of images (YOLOv8 RGB) gave better

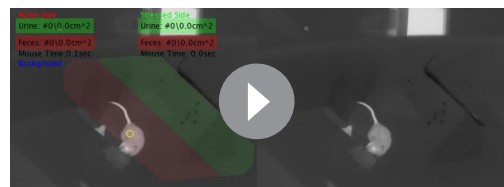

**Video 1.** IR video of a single ESPs trial of a male mouse with an overlay of the automatic detections. Automatic detections are overlayed in red for feces, green for urine, and blue for BG. The stressed mouse side of the arena is marked in green, and the object side is marked in red. Counters of the number and area of automatic detections in each side of the arena are written on the top left. The video plays at X8 speed.

https://elifesciences.org/articles/100739/figures#video1

results compared to a single one (YOLOv8 Gray) suggests that temporal information is important for the accurate detection and classification of deposition events.

Finally, to test the accuracy of DeePosit across different mouse strains and experimental arenas, we evaluated DeePosit accuracy for SP and SxP tests performed by C57BL/6 black mice ($n$ = 10) in a white Plexiglass arena. DeePosit achieved good performance (mean F1 = 0.81), even though videos with black mice or with white arenas were not included in the training set (see *Figure 3—figure supplement 1*). Thus, DeePosit shows stable accuracy across experimental conditions.

Our code allows changing the main parameters of the algorithm in order to adjust them to the relevant settings. Therefore, we examined the sensitivity of DeePosit to changes in the parameters used by the algorithm. We first examined DeePosit accuracy as a function of the $\Delta T_{Threshold}$ parameter of the preliminary heuristic detection (see methods). We found that $\Delta T_{Threshold}$ = 1.6°C gave the best performance in our setting (see *Figure 3—figure supplement 2*), although the accuracy was quite stable (mean F1 score of 0.88–0.89) for values between 1.1 and 3°C. We also trained the DeePosit classifier with an input time window of [–11..30] s instead of [–11..60] s and got no difference in the accuracy level (mean F1 score of 0.89 in both cases).

## Distinct dynamics of urination and defecation activities across the various tests

*Figure 4a, b* shows the raw results of urine and fecal deposit detection by the DeePosit algorithm as a function of time across all three tests, for each male (blue symbols) and female (red symbols) subject mouse. The symbols representing the various deposit types are also labeled (with black dots) according to the arena side of each deposition (relative to the two stimuli). These raw results were further analyzed by computing the average number of urine or fecal deposits, per minute (*Figure 4C*). The area of the deposits (cm$^2$) is also plotted (*Figure 4d*), since urine deposit size might vary significantly between distinct events and conditions (*Wegner et al., 2018*). In general, the event rate and deposit area showed similar trends. As for the side preference, females showed a slight tendency to a higher urination rate at the social stimulus side in the SP test, while males showed a tendency to a higher defecation rate at the social stimulus side (see *Figure 4—figure supplement 2*). Importantly, urination and defecation activities showed distinct dynamics from each other: defecation exhibited a single clear peak in an early stage of the habituation, which appeared in all cases. In contrast, urination was characterized by two peaks, which were not visible in the SP test but appeared in the SxP and got even stronger in the ESPs test, thus showing a gradual increase across test days. The first urination peak occurred in males at the early habituation stage, parallel to the peak in defecation, while the second urination peak occurred in both males and females at the beginning of the trial stage, after stimuli insertion into the arena. For statistical analysis of these dynamics, we compared the mean urine and fecal deposition rates between three periods: the beginning of habituation (habituation minutes 1–4), the end of habituation (habituation minutes 11–14), and the trial - after stimuli introduction (trial minutes 1–4) (*Figure 5a, b*). The last minute of both the habituation and the trial stages was not included in the analysis since DeePosit uses 1 min of video after the deposition as input; hence, the accuracy may be lower in cases where we have less than 1 min of video after the deposition. However, including the missing minute of each stage in the analysis yielded similar results (see

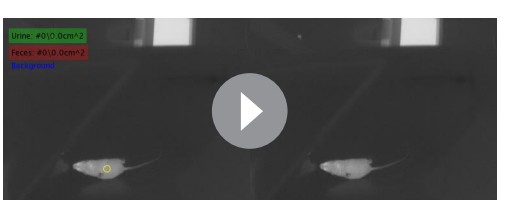

**Video 2.** IR video of a single ESP habituation of a male mouse with an overlay of the automatic detections. The video shows the habituation part of the experiment in *Video 1*.

https://elifesciences.org/articles/100739/figures#video2

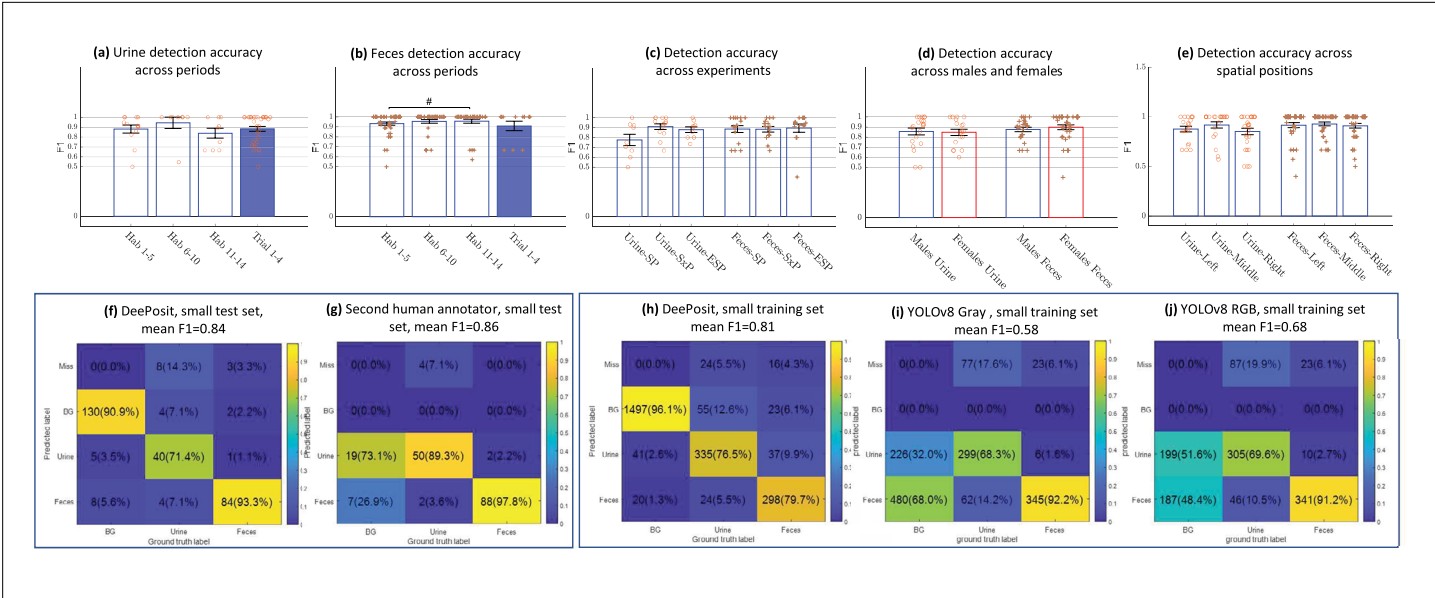

**Figure 3.** Validation of DeePosit accuracy. Mean accuracy ± SEM of urine (**a**) and fecal (**b**) deposits detection by DeePosit, as measured by F1 score across various stages of the experiment. Each '+' or 'o' marks the F1 accuracy for a single mouse in a single experiment. No significant difference was found. Similarly, DeePosit accuracy was not significantly affected by the experiment type (**c**), by the sex of the subject mouse (**d**), or by the spatial location of the deposition in the arena (arena's floor was divided into three equal parts) (**e**). A two-sided Wilcoxon rank sum test was used. (**a–c, e**) are FDR corrected rank sum tests (**Benjamini and Hochberg, 1995**). The # at (**b**) stands for FDR corrected p-value of 0.08. Sixty test videos (24 videos with a male subject mouse and 36 with a female) were used in (**a, b, d**). Forty-six test videos were used in (**c, e**) of which 18, 14, 14 videos were SP, SxP, and ESPs accordingly. Mice without manually annotated depositions of the relevant type (either urine or feces) during the relevant period, experiment, or spatial location were ignored (since F1 is not defined in such cases). Since differentiating small urine and feces in thermal videos can be a challenging task even for humans, we evaluated the accuracy of a second human annotator on 25 test videos of CD1 mice (a subset of the full test set) and reported both the accuracy achieved by DeePosit (**f**) and the second human annotator (**g**) on these test videos. The mean F1 score, $(F1_{Urine} + F1_{Feces})/2$ is 0.86 for the second human annotator and 0.84 for the DeePosit algorithm. To compare our result with another popular object detection approach, we annotated 39 training videos of CD1 mice with bounding boxes to match the YOLOv8 framework. For fairness, we trained both algorithms on the same training set of videos. We tested the accuracy on the test set which includes 60 videos. (**h**) shows the confusion matrix for DeePosit, while (**i, j**) show the confusion matrices achieved using YOLOv8 with a single image as input (YOLOv8 Gray) and with three images as input representing time $t + 0$, $t + 10$, $t + 30$ s from each event (YOLOv8 RGB). DeePosit accuracy surpasses YOLOv8 results in both cases. YOLOv8 RGB accuracy surpasses YOLOv8 Gray, suggesting that temporal information is helpful in the detection of urine and feces.

The online version of this article includes the following figure supplement(s) for figure 3:

**Figure supplement 1.** Accuracy for small and large detections in C57BL/6 mice.

**Figure supplement 2.** Detection accuracy at various values of $\Delta T_{Threshold}$.

**Figure supplement 3.** Examples of detections in test videos.

*Figure 5—figure supplement 1*). For both males and females and across all tests (besides female SP, where only a trend was observed), we found a significantly higher level of fecal deposition at the beginning of habituation than at the habituation end and the trial stage. In contrast, a similar comparison of urination showed that its level was significantly higher during early habituation than at the end of it only for males in the SxP and ESPs tests. A similar elevation in urination was observed during the trial stage, as compared to the habituation end, for both males and females, again specifically during the SxP and ESPs tests. Interestingly, we found an opposite trend for fecal deposits, with a significant decrease in defecation rate during the trial, as compared to the end of habituation, in all the tests for males and in the SxP test for females (*Figure 5a, b*). Similar results were found for urine and fecal deposit areas (*Figure 5—figure supplement 2*). Moreover, similar trends were observed when the proportion of mice actively depositing urine or feces during each stage was calculated for each case (*Figure 5c*). These data reveal distinct dynamics for urination and defecation activities in a sex- and test-specific manner.

When comparing the urination and defecation patterns of CD1 male mice with those observed in C57BL/6 male mice (*Figure 4—figure supplement 1*, *Figure 5d*, and *Figure 5—figure supplement*

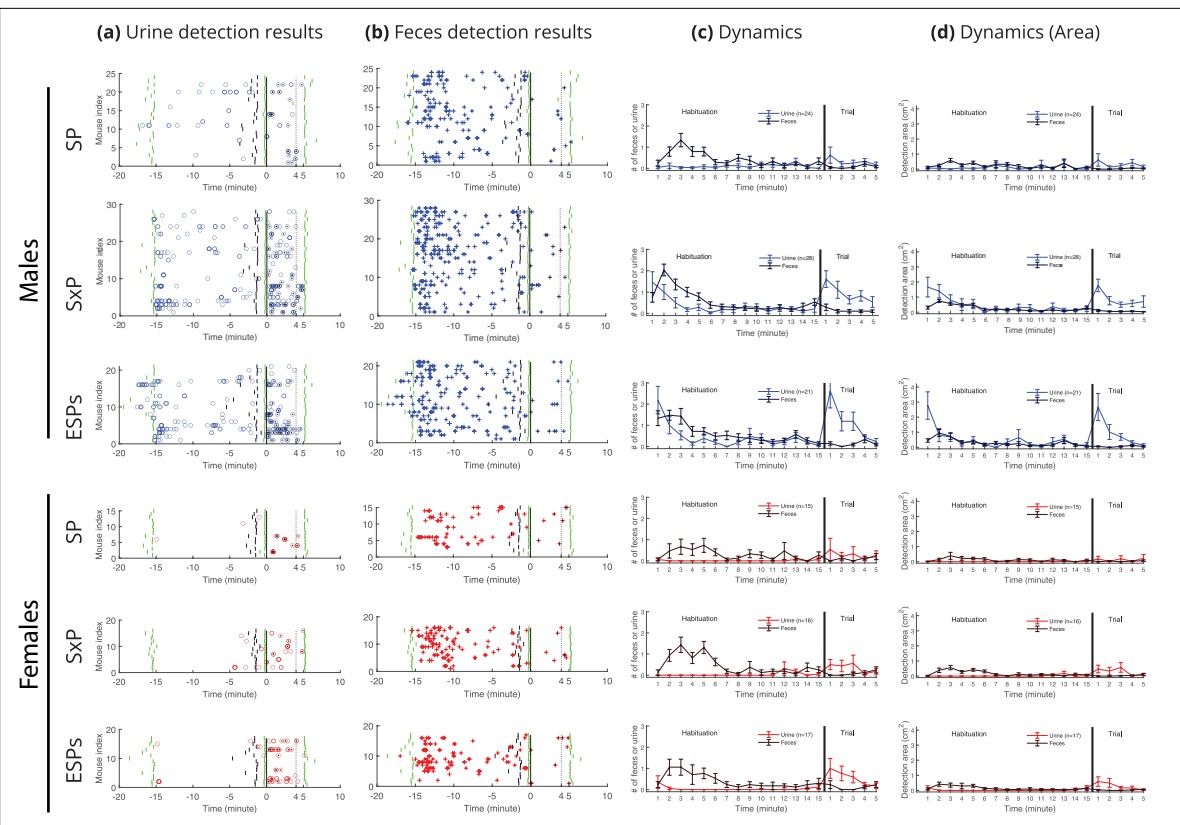

**Figure 4.** Urine and fecal deposition detection results across tests in CD1 mice. Each o represents a single detection of urine deposition (**a**), while each + represents a single detection of fecal deposition (**b**). A black dot in the center of a circle or a + sign marks that this detection is on the side of the preferred stimulus, defined as the social stimulus in the SP trial, the female in the SxP trial, and the stressed mouse in the ESPs trial. Short green lines mark the start and end of the habituation stage and the end of the trial stage, while short vertical black lines mark the end of minute 14 of the habituation stage. The vertical black line at time = 0 marks the start of the trial stage after stimuli introduction to the arena, while the vertical dashed line marks 4 min after the beginning of the trial. Dynamics plots (right) show mean rate (**c**) and mean area (**d**) per minute for both urine and fecal deposits. Error bars represent ± SEM.

The online version of this article includes the following figure supplement(s) for figure 4:

**Figure supplement 1.** Urine and fecal deposition detection results across tests in C57BL/6 mice.

**Figure supplement 2.** Urine and fecal deposition side preference.

*1d*), we found distinct characteristics. In contrast to CD1 mice, urination rate of C57BL/6 mice was higher at the beginning of habituation compared to the end of it already in the SP experiment. On the other hand, urination rate of C57BL/6 mice did not increase during the trial as compared to the end of habituation in any of the experiments. Notably, unlike CD1 mice, of C57BL/6 mice did not deposit urine spots smaller than 1 cm$^2$ (compare *Figure 3—figure supplement 1* with *Figure 2—figure supplement 1*). As for the defecation rate of C57BL/6 mice, similarly to CD1 mice, it was higher at the beginning of habituation compared to the end of it. However, unlike the trend in CD1 mice, it was not reduced in the trial stage, as compared to end of habituation. Thus, the distinct dynamics of urination and defecation activities observed using DeePosit, are mouse strain specific.

## Sex-dependent differences across the various stages

We used two types of statistical tests to compare between male and female CD1 mice. A two-sided Wilcoxon rank sum test (significance marked by *) was used for all pairwise comparisons. In addition, since some of the data was zero-inflated (many mice did not deposit urine or feces at all during the relevant period), we used a two-way Chi-square test (significance marked by +) to compare the distribution of zeros and non-zeros in the male group versus the female group. A test-dependent significant difference between males and females was found in the early stage of

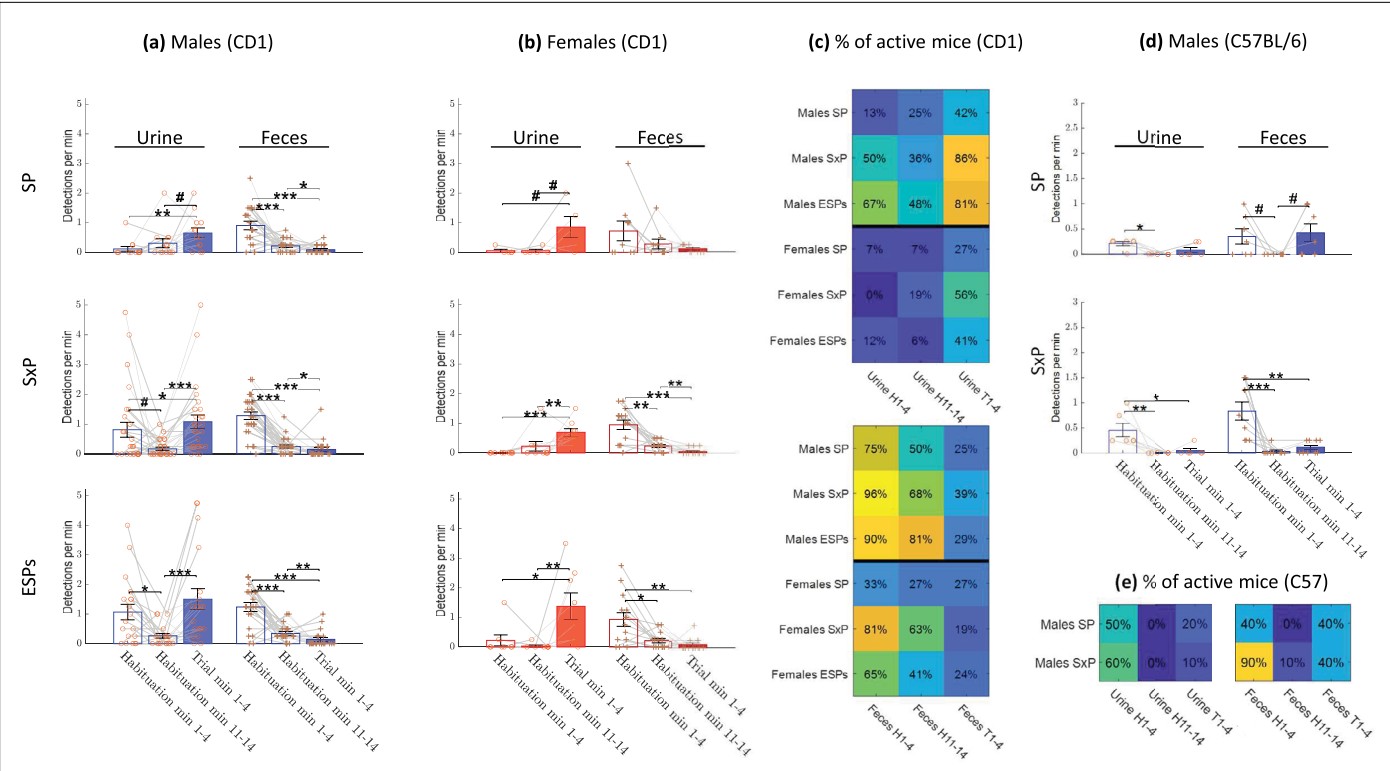

**Figure 5.** Comparison between test stages. Mean rate ± SEM of urination and defecation events detected during habituation start (minutes 1–4), habituation end (minutes 11–14), and trial (minutes 1–4) stages, for male CD1 mice (**a**), female CD1 mice (**b**) and male C57BL/6 mice (**d**). Percent of active mice (mice with at least one detection) across tests during habituation start, habituation end, and trial stages, for CD1 mice (**c**) and for male C57BL/6 mice (**e**). Two-sided Wilcoxon rank sum test equal to or smaller than 0.1, 0.05, 0.01, and 0.001 was marked with #, *, **, and ***, respectively. In (**a, b, d**), only mice with urination in at least one of the periods were included in the urine analysis. Same for feces. In (**a, b**), n = 13, 27, 19 male CD1 urination in SP, SxP, and ESPs, and n = 21, 28, 21 for defecation. Accordingly, for CD1 females, n = 5, 9, 8 for urination and n = 9, 14, 14 for defecation. In (**d**), n = 6, 6 for urination and n = 7, 9 defecation in SP and SxP. In (**c**), the total number of CD1 male mice is 24, 28, 21 in SP, SxP, and ESPs, and the total number of female mice is 15, 16, 17. In (**e**), the total number of male C57BL/6 mice is 10, 10 in SP and SxP.

The online version of this article includes the following figure supplement(s) for figure 5:

**Figure supplement 1.** Comparison of deposition events rate between test stages using 5-min periods.

**Figure supplement 2.** Comparison of deposition area between test stages using 4-min periods.

habituation (*Figure 6a*). On the first day of experiments (the SP test), males and females showed a low urination rate at the first 4 min of habituation, with no significant difference between them. However, in the next two testing days (SxP and ESPs tests), when the mice were already familiar with the arena, we found a significantly higher rate and area of urine deposition in males compared to females (*Figure 6a*, *Figure 6—figure supplement 1a*). As for defecation events, males showed a significantly higher level in this period, in all tests. During the last stage of habituation (minutes 11–14), we found a significant difference between males and females only for the ESPs test, with males showing higher levels of both urination and defecation rate (*Figure 6b*) and area (*Figure 6—figure supplement 1b*).

For statistical comparison between males and females during the trial, where an initial peak was observed in some cases (*Figure 4c, d*), we divided the trial stage into two periods: the first minute and minutes 2–4, and averaged the results of each period separately. As apparent in *Figure 6c, d*, *Figure 6—figure supplement 1c, d*, the urination rate during the first minute of the trial stage showed no sex-dependent difference in the SP test. In contrast, a significantly higher level was observed for males versus females in the SxP and ESPs tests. No sex-dependent difference in urination rate was observed For trial minutes 2–4, or in defecation rate for any of the trial periods.

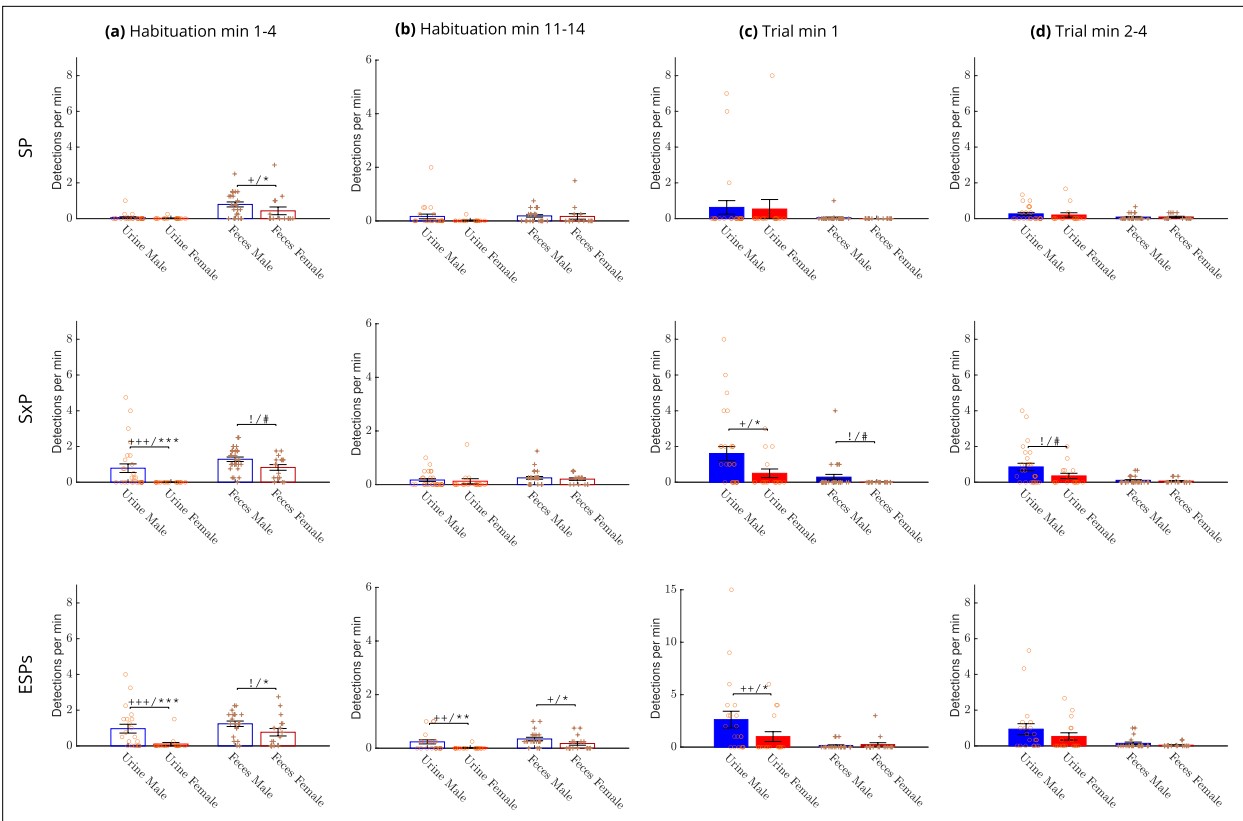

**Figure 6.** Comparison of deposition rates between sexes. The mean rate ± SEM of urination and defecation events for males (blue bars) versus females (red bars) during early (minutes 1–4) (**a**) and late (minutes 11–14) (**b**) periods of the habituation stage and during the first minute (**c**) and minutes 2–4 of the trial stage (**d**) . A significant difference between the mean rate of urine or fecal depositions (two sided Wilcoxon rank sum test) with p value equal to or smaller than 0.1, 0.05, 0.01, 0.001 was marked with #, *, **, ***, respectively. A significant difference in the distribution of non-depositing animals (Chi-square test) with p value equal to or smaller than 0.1, 0.05, 0.01, 0.001 was marked with !,+,++,+++ respectively. For male mice, $n$ = 24, 28, 21 for SP, SxP, and ESPs. For female mice, $n$ = 15, 16, 17 accordingly.

The online version of this article includes the following figure supplement(s) for figure 6:

**Figure supplement 1.** Comparison of mean deposition areas between sexes.

## Male urine and fecal deposition rates are test dependent

Since the data so far suggest a dynamic change from the SP (first day) to the SxP (second day) and ESPs (third day) tests specifically for males, we checked the effect of test type (SP, SxP, and ESPs) on the dynamics of urination and defecation activities using Kruskal–Wallis test (*Table 1* and *Appendix 1—table 1*). The urination and defecation rates (*Table 1*) and deposits areas (*Appendix 1—table 1*) of males showed both a significant effect of the test type, with urination showing this effect during early

**Table 1.** The effect of the test (SP, SxP, and ESPs) on urination or defecation events rates.

Kruskal–Wallis test was used to check if the test type affects the rate of urination or defecation events. p-value equal to or smaller than 0.1, 0.05, 0.01, 0.001 was marked with #, *, **, ***, respectively. For male mice, $n$ = 24, 28, 21 for SP, SxP, and ESPs. For female mice, $n$ = 15, 16, 17 accordingly.

| Measurement | Habituation, minutes 1–4 | Habituation, minutes 11–14 | Trial, minute 1 | Trial, minutes 2–4 |
|---|---|---|---|---|
| Male #Urine | 0.0004*** | 0.3804 | 0.0015** | 0.0301* |
| Female #Urine | 0.3777 | 0.3943 | 0.4287 | 0.3918 |
| Male #Feces | 0.0221* | 0.1178 | 0.3054 | 0.9251 |
| Female #Feces | 0.0635# | 0.2653 | 0.1553 | 0.5663 |

habituation and during the trial, while defecation showing such effect at early habituation, but not during the trial stage. No significant effect was found for females.

## Discussion

Here, we present a new algorithm and an open-code trainable AI-based computational tool for detecting and classifying urination and defecation events from thermal video clips. This algorithm enables a detailed characterization of the dynamics of urination and defecation activities during social behavior of small rodents. One advantage of this tool is that it is automated, thus allowing a rapid and observer-unbiased analysis of urine and fecal deposition events and areas, with a good temporal and spatial resolution. Specifically, combining our algorithm with an IR camera for thermal imaging of behavioral experiments can replace the void spot test, which usually lacks any temporal resolution and is prone to mistakes caused by urine smearing and filter-paper tearing. Finally, our algorithm facilitates the analysis of defecation activity, which was rather unexplored so far but may contribute to scent-marking behavior, as discussed below. Our algorithm uses thermal video clips generated by an IR camera placed above the arena and does not require a camera placed below a clear arena floor, as used by a recent paper (see *Keller et al., 2018* for example). Thus, it can be utilized for analyzing experiments conducted in standard experimental setups, such as those used for the three-chamber test. The computational tool and experimental method presented here may be useful for a detailed characterization of social behavior in mice, including murine models of autism spectrum disorder and other pathological conditions. It may also be used to explore urination and defecation activities in other scientific contexts, unrelated to social behavior. Finally, our experimental setup is cheap and easy to assemble, and the detection algorithm can run on a standard PC with a GPU card.

Analysis of the errors made by the algorithm in the test dataset (see *Figure 2—video 1* for video clips of these events) raised several limitations, that might be addressed in future work. Urine or fecal deposits must be fully visible while the deposit is still warm. A close adjacency between the mouse and the deposit might cause the mouse mask to overlap the mask of the deposit, thus preventing its detection. Many of the 'miss' events in the test video clips were created by the mouse staying close to the urine or fecal deposits for a long period after their deposition. Few other 'miss' events were due to very small urine spots or due to repeated urination in the same position during a very short time period, which resulted in detecting these separate urination events as a single event by the algorithm. A wrong classification of urine as fecal deposition occurred in 2.3% of the urination events. In many of these events, the urination spot was small (and therefore harder to distinguish from a fecal deposition) (see *Figure 3—figure supplement 3c*). Wrong classification of background as feces occurred 21 times in the test set. In most of these events, the mistake was due to feces that were moved by the mouse to a new location while still being warm. Such cases may be mitigated in future work by a tracking algorithm that continuously tracks the location of each fecal deposit. Wrong classification of background as urine occurred 33 times in the test set, with some of these errors caused by smearing of large warm urine spots.

We evaluated the accuracy of the algorithm and found it to be uniform across the various sexes, tests and session stages of the experiments used by us. This suggests that the low level of errors made by the algorithm should not create a bias during biological experiments. Moreover, the algorithm achieved a good and stable accuracy even for C57BL/6 mice examined in a while arena, a condition that was not represented in the training videos. Thus, the algorithm seems to be robust, with a low sensitivity to changing conditions. We also compared the algorithm's accuracy to the accuracy achieved by a second human annotator on the same dataset and concluded that the algorithm accuracy is comparable to the accuracy of a human annotator while being much faster and unbiased. Finally, the algorithm showed superior performance over classic object detection algorithms, such as YOLOv8, which are based on a single image input. This is most likely due to the transformer-based architecture of our algorithm, which allows it to use the temporal information extracted from the thermal video clips.

Future work might improve DeePosit by extending the training set and including more challenging examples. Notably, comparing a small training set (*Figure 3h*) with a larger one (*Figure 2h*) shows that the larger training set improved the accuracy of DeePosit. Another way for future improvement in DeePosit accuracy may be by using a trainable detection and segmentation algorithm instead of heuristic preliminary detection. Note that our classifier currently does not get the mask of the

preliminary detection as an input, making the classification task harder when there are adjacent deposition events. An end-to-end trainable detection, segmentation, and classification pipeline might address these limitations but will require a much larger training set. Future work might also adapt the algorithm for multi-animal experiments. Such adaptation might require detecting the mask of each of the animals, separating the identity of each of the animals, and associating each deposition with the relevant animal.

We validated our method and algorithm using experimental results from social discrimination tests conducted by male and female CD1 and male C57BL/6 mice. We demonstrated distinct dynamics of urination and defecation activities across the habituation and trial stages, with sex-, test-, and strain-dependent differences. Both male and female CD1 mice, as well as male C57BL/6 mice showed higher rates of defecation activity at the early stage of the habituation phase, as compared to later stages (*Figure 5*). This tendency may reflect a higher level of anxiety at the beginning of the habituation phase, caused by the novel context. Still, it may also serve for scent-marking activity, that labels the arena as a familiar environment. The latter explanation is supported by the fact that the peak in defecation activity was not reduced from the first-day test (SP) to the second and third-day tests (SxP and ESPs), when the subject is expected to be less anxious due to the familiar context. In contrast to defecation, urination activity at the beginning of the habitation phase in CD1 mice was test dependent. While no peak was observed during the SP test, the first time the animals were exposed to the experimental arena, it was observed in the second test (SxP) and got even stronger in the last test (ESPs). This development was statistically significant in CD1 males but not in females. Since these changes occur during the habituation phase, before the introduction of stimuli to the arena, they cannot reflect the type of test and thus seem to be induced by the order of the experiments. Notably, similar dynamics across experimental days were previously reported using the void spot assay for C57BL/6j mice (*Keil et al., 2016*). This suggests that the induction of urination activity by males at the early stage of the habituation phase represents territorial scent-marking activity, which is positively correlated with the higher familiarity experienced by the subject in the arena as the experiments progressed between days. It should be noted that an early peak of urination upon entering an environment was reported by a recent study using a thermal camera for manual analysis of urination activity (*Miller et al., 2023b*). A second peak of urination activity was observed at the beginning of the trial period, after stimuli insertion to the arena. This was observed in both male and female CD1 mice, but the test type significantly affected it only in males. In this case, we cannot dissect the effect of test type from the test order, as the urination activity occurred after stimuli insertion and, hence, may be induced by the presence of specific social stimuli. Since the subjects are already habituated to the arena at this stage, the elevated urination activity seems to serve as part of the subjects' social behavior, most probably as a territorial scent-marking behavior induced by the presence of social stimuli, i.e., competitors. Interestingly, we found several differences in the dynamics of CD1 male mice and C57BL/6 male mice, suggesting that the scent-marking behavior is also strain specific. Unlike CD1 male mice, C57BL/6 male mice exhibited a peak in urination already at the beginning of the first (SP) habituation, a trend toward higher level of defecation activity in the SP trial stage, and no increase in urination activity during the SP and SxP trial stage, compared to the habituation end. However, several findings were common for both CD1 and C57BL/6 male mice, such as the higher feces rate at the beginning of habituation in comparison to the end of habituation and the higher levels of urination at the beginning of the SxP habituation stage.

We did not observe a consistent spatial distribution of the urine or fecal deposits between the arena sides of the preferred and non-preferred stimuli in CD1 mice. This seems to contradict a recent study (*Miller et al., 2023b*), that reported opposite bias toward familiar versus unfamiliar stimuli in losers versus winners wild-derived mice following a social contest. This contradiction may be due to the distinct mouse strains or the distinct contexts of social behavior (presentation of a single stimulus animal in comparison to two simultaneously presented animals) used by both studies.

Overall, the novel algorithm and software presented here enable a cost-effective, rapid, and unbiased analysis of urination and defecation activities of behaving mice from thermal video clips. The algorithm is trainable and may be adapted to various behavioral and experimental contexts. Thus, it may pave the way for the integration of this important behavioral aspect in the analysis of small rodents' social and non-social behaviors, in health and disease.

# Materials and methods

## Animals

Subject animals were adult (12–14 weeks old) male and female wild-type offspring derived from breeding couples of *Gtf2i+/Dup* with a CD1 (ICR) genetic background mice (*Mervis et al., 2012*). We used this line of mice since, in parallel to this work, we are investigating the phenotype of the *Gtf2i+/Dup* mutant line. These results will be published separately. These mice were bred and grown in the SPF mouse facility of the University of Haifa. C57BL/6 mice were purchased from Envigo (Rehovot, Israel). Stimulus animals were adult (12–14 weeks old) male and female CD1 or C57BL/6 mice purchased from Envigo (Rehovot, Israel). All mice were housed in groups of 3–5 in a dark/light 12 hr cycle (lights on at 7 pm), with ad libitum food and water under veterinary inspection. Experiments were performed in the dark phase of the dark/light cycle. All experiments were approved by the University of Haifa ethics committee (Reference #: UoH-IL-2301-103-4).

## Setup and video acquisition

The experimental setup is based on the setup described in *Netser et al., 2019*. Briefly, a black or white Plexiglass box arena (37 cm × 22 cm × 35 cm) was placed in a sound-attenuated chamber. A visible light (VIS) camera (both Flea3 and Grasshopper3 models manufactured by Teledyne FLIR were used, both with a wide-angle lens, rate of 30 frames per second, and USB3 interface) and a long wave IR camera (Opgal's Thermapp MD with 6.8 mm lens, 384 × 288 pixels at a rate of 8.66 frames per second (FPS)) were placed about 70 cm above the arena's floor. The IR camera was designed to measure human skin temperature and outputs the apparent temperature for each pixel. Raw pixel values were converted to Celsius degrees using the formula supplied by the manufacturer. We acquired the camera videos using custom-made Python software (code is available at https://github.com/davidpl2/DeePosit [copy archived at *Peles, 2025*] and https://doi.org/10.5281/zenodo.14754159) that used the manufacturer's SDK (SDK version: EyeR-op-SDK-x86-64-2.15.915.8688-MD). To improve the accuracy of and reduce possible drifts in the measured temperature, a high-emissivity blackbody (Nightingale BTR-03 blackbody by Santa Barbara Infrared, Inc) was placed in the camera's field of view and was set to 37°C. During analysis, the offset between the blackbody apparent temperature and 37°C was subtracted from the image. To improve image quality, we turned on the camera at least 15 min before the beginning of the experiment (this allows the camera's temperature to get stable). In addition, to reduce pixel non-uniformity, we captured 16 frames of a uniform surface (a piece of cardboard placed in front of the camera) before each test. These images were then averaged, and the average image's mean was subtracted from it to get a non-uniformity image with zero mean. The non-uniformity image was then subtracted from each image in the video to achieve better pixel uniformity.

## Social behavior tests

We used three distinct social discrimination tests, as previously described in *Mohapatra et al., 2024*. Briefly, all tests consisted of 15 min of habituation, during which the subject mouse got used to the arena with empty triangular chambers (12 cm isosceles, 35 cm height) located at randomly chosen opposite corners. Each triangular chamber had a metal mesh (18 mm × 6 cm; 1 cm × 1 cm holes) at its bottom, through which subject mice could interact with the stimuli. After habituation, the empty chambers were removed and new stimuli-containing chambers were introduced into the arena for the 5-min trial. In the SP test, a novel (i.e., unfamiliar to the subject mouse) sex-matched stimulus mouse was placed in one chamber, whereas an object stimulus (a Lego toy) was placed in the opposite chamber. In the Sex Preference (SxP) test, a novel female mouse was placed in one chamber while a novel male was placed in the opposite chamber. In the ESPs test, a novel stressed (restrained in a 50-ml plastic tube for 15 min before the test) sex-matched mouse was introduced to one chamber of the arena while a novel naïve mouse was placed in the opposite chamber.

## Behavioral analysis

VIS video clips were analyzed using TrackRodent (*Netser, 2020*), as previously described in *Netser et al., 2017*.

## Urine and feces detection algorithm

The detection algorithm consists of two main parts. A preliminary heuristic detection algorithm detects warm blobs. These blobs are then fed into a machine learning-based classifier, which classifies them as either urine, feces, or background (i.e., no detection). The algorithm's code is available here: https://github.com/davidpl2/DeePosit and https://doi.org/10.5281/zenodo.14754159.

### Manual inputs

A graphical user interface was developed in Matlab to support all of the required manual annotations. Each video went through a manual annotation of the arena's floor, the area of the blackbody, and a specification of the first and last frames of both the habituation and trial periods. These two periods were separated by a ~30-s period during which the stimuli were introduced to the arena, which was excluded from the analysis. Also, the arena side of each stimulus (e.g., the male and female sides in the SxP test) was defined as the half of the arena close to this stimulus's chamber. To generate the train and test sets, a human annotator manually tagged urine and fecal deposition events in videos of 157 experiments with CD1 mice, of which 97 were used for training and 60 for testing. A single click was used to mark the center of each urine or fecal deposit in the first frame where it was clearly visible. The training set included 751 urine annotations and 637 feces annotations. The test set included 438 urine annotations and 374 feces annotations. Additional details can be found in the software's manual.

### Preliminary detection of hot blobs

Urine and fecal deposits appear as hot (bright) blobs in the first seconds after deposition. After a cool-down period, which takes about 30–60 s for feces and small urine spots and up to ~4 min for large urine spots, feces and urine appear as dark spots in the thermal image. The preliminary detection relies on these effects (see pseudo-code in Algorithm 1). It uses image subtraction to search for hot blobs that appear in the video and cool down later. We generate a background image $B_i$ for each frame $F_i$ to detect new hot blobs. Subtraction of $B_i$ from $F_i$ generates an image in which the mouse pixels and new (warm) urine and feces pixels appear bright. We set $B_0$ as the per-pixel minimum of the first 20 s of video (note that habituation and trial videos are analyzed separately to account for possible minor shifts in the arena's position). We assume that the mouse is brighter than the arena's floor and that the mouse moves during the first 20 s, so each pixel will get the arena's floor value at least once during this time.

For $i > 0$ we compute $B_i$ as the minimum of images $N_j$, $j \in [i-44, .., i-36]$ (this roughly matches time range $[i-5s, .., i-4s]$) where $N_j$ is an image in which the mouse pixels were replaced by the last known values from before the time that the mouse occupied these pixels. We set $N_{j \leq 0} = B_0$.

To compute the mouse mask at frame i, $B_{i-1}$ is subtracted from $F_i$. The subtraction result is dilated by Matlab's *imdilate* function with a structuring element of a disk of a radius of 2 pixels and then compared against a threshold of 1°C to get a binary mask of the pixels that are warmer than the arena's floor. Connected regions are then computed using Matlab's *bwlabel* function and the connected region with the largest intersection with the arena's floor is considered as the mask of the mouse (denoted $M_i$).

$N_i$ is then computed by taking $F_i$ values for the pixels outside $M_i$ and taking the values of $N_{i-1}$ for the mouse containing pixels: $N_i = N_{i-1} * M_i + F_i * (1 - M_i)$ where * denotes pixel-wise multiplication.

The difference image $D_i$ is computed by: $D_i = F_i - max(T, B_i)$ where $T$ is the arena's floor median temperature, computed by $T = median(B_i(AF \& \neg M_i \& \neg M_{i-1}))$ where $AF$ is a mask of the arena's floor, & is pixel-wise AND operation, and $\neg$ is pixel-wise NOT operations. Using $T$ prevents higher detection sensitivity in darker regions of the arena floor (regions in the arena's floor that are covered in cooled-down urine appear darker than dry regions of the arena's floor, see *Figure 2e*).

The cooldown rate $CD_i$ is computed by taking the per pixel minimum of the frames in the next 40 s following $F_i$ and subtracting it from $F_i$.

The hot blobs mask $BM_i$ is computed by taking the pixels for which $D_i > \Delta T_{Threshold}$ and not included in $M_i$ and $M_{i-1}$ and for which the $CD_i > 1.1$ and $CD_i > 0.5 * D_i$. We explored several values for $\Delta T_{Threshold}$ (see *Figure 3—figure supplement 2*) and chose $\Delta T_{Threshold} = 1.6°C$ as the default value for this parameter. We ask for the cooldown to be at least half of the increase in the temperature but not more than that since very large urinations cool down slower and might take more than 40 s to cool

down fully. We excluded pixels in $M_{i-1}$ (mouse containing pixels in frame i-1) and not just $M_i$ since the IR sensor has a response time that might causes pixels included in $M_{i-1}$ to be slightly brighter.

$BM_i$ goes through a morphological close operation using Matlab's *imclose* function with a structure element of a disk with a radius of 4 pixels. This causes any nearby drops of urine to unify to a single detection. Blobs that overlap pixels outside the arena's floor or touch the mouse mask are ignored to avoid detection on darker areas of the mouse (mostly the tail), reflections from the arena's wall, and detections due to a stimulus mouse which sometimes sticks his nose throughout the barrier net of the chamber. Also, blobs with a size <2 pixels or larger than 900 pixels are ignored (pixel size is roughly 0.02 cm$^2$).

Blobs that intersect previously detected blobs are considered to be the same detection if no more than 30 s passed from the last frame in which the previous detection was last detected. A unified detection mask is computed each time a detection is associated with a previous detection. This allows reduction of false alarms which might be caused by the smearing of a still-hot urine drop. If no such intersection exists, a new preliminary detection is added to the list of detections. A blob should be detected in at least two frames to be included in the output detections. The selected frame ID for each blob is the frame that contains the maximum intensity for this blob out of all frames in which this blob was detected. The representative coordinates for each detected blob were chosen by taking the pixel with the maximum intensity inside the blob in the selected frame. Usually, the selected frame for each blob is the first frame of the detection (as the detection cools, the maximum intensity is usually in the first detected frame). Still, it might be another frame if the detection was partly occluded by the mouse tail or if a second urine event occurred in the same place during the relevant time frame. The output detections are fed into a classifier, which will be described next.

The detection threshold $\Delta T_{Threshold}$ is higher than the mouse detection threshold (1°C) to avoid false defections within the borders of the subject mouse body.

---

Algorithm 1 Preliminary Detection of Hot Blobs

---

$B_0(p) \leftarrow min_{i\in[1..20FPS]}(F_i(p))$ ▷ Background image at pixel p.
$N_{i<=0} \leftarrow B_0$
Let $F_i$ be the i'th frame in the video
Let $AF$ be the mask of the arena's floor (equals 1 for the arena's floor pixels and 0 elsewhere)
**for** $i \in [1..n]$ **do**
 $M_i \leftarrow Blob\_With\_Maximal\_Floor\_Intersection(imdilate((F_i - B_{i-1}) > 1°C, radius = 2))$
 $N_i \leftarrow N_{i-1}M_i + F_i(1 - M_i)$ ▷ Mouse pixels are replaced with background pixels
 $B_i(p) \leftarrow min_{j\in[i-5sec,...,i-4sec]}N_j(p)$ ▷ Background value for each pixel p
 $T \leftarrow median(B_i(AF\&\neg M_i\&\neg M_{i-1}))$ ▷ Median temperature of the arena floor
 $D_i \leftarrow F_i - max(T, B_i)$ ▷ Difference image. max operation is pixel-wise
 $CD_i(p) \leftarrow F_i(p) - min_{j\in[i..min(n,i+40*FPS)]}F_j(p)$ ▷ Cooldown in the next 40 sec
 $BM_i \leftarrow (D_i > \Delta T_{Threshold})\&\neg M_i\&\neg M_{i-1}\&(CD_i > 1.1°C)\&(CD_i > 0.5 * D_i)$ ▷ Hot Blobs Mask
 $BM_i \leftarrow imclose(BM_i, radius = 4)$ ▷ Filling small gaps in blobs mask
 that are fully inside $AF$ and not adjacent to mouse's mask
 with $size \in [2..900]$
 blobsList is updated. New blobs are associated with blobs that were detected up to
 30 seconds ago if their masks intersect
**end for**
return blobs in blosbList that were detected in at least 2 frames

---

## Classifying preliminary detections using an artificial neural network

Preliminary detections are fed to a trained artificial neural network classifier which classifies them as either: *Urine*, *Feces*, or *Background* (**Figure 2g**). We relied on the transformer-based architecture proposed by **Carion et al., 2020**. This architecture was designed for object detection in RGB images. It receives an RGB image as input and outputs a set of bounding boxes around each detected object and the classification of each detection. In brief, this neural network architecture consists of a convolutional neural network based on the *ResNet* architecture proposed by **He et al., 2016**, which serves as the backbone and extracts a set of feature vectors from each location in the input image. The feature vectors are attached with a position encoding, which is a second feature vector that describes the spatial location in the input image, associated with the backbone's feature vector. For each spatial location, the feature vectors from the backbone and the positional encoding are summed and fed into an encoder transformer, which uses an attention mechanism to share information between the feature vectors from various spatial locations. A decoder block is fed with

the output of the encoder, and an additional set of vectors is denoted as queries. The decoder uses several layers of self and cross-attention to share information between queries (self-attention) and between the queries and the decoder output (cross-attention). Finally, the encoder outputs a feature vector for each input query. This vector is fed into a feed-forward network to compute each query's bounding box and classification. One of the possible classification outputs for each query is 'no object'. We relied on the popular open-source code published by *Carion et al., 2020* and made a few adjustments. Instead of feeding a single RGB image as input, for each detection in $F_i$ we used a series of 78 grayscale image patches cropped around the detection pixel (65 × 65 pixels patch) and representing a time window of about [−11s .. 60s] around the detection. For detection in $F_i$ we used the frames $[F_{i-12*8}, F_{i-11*8}, ...F_{i-0*8}, ..., F_{i+65*8}]$ for classification. We used this relatively large time window to capture the cooldown of the feces and urine, movement of feces (which are frequently moved by the mouse), or smearing of urine. Additionally, this time window allows for capturing the moment of the deposition of the urine or feces, which sometimes occurs a few seconds before the preliminary detection (since the mouse may fully or partly occlude the detection in the first seconds). In case one or more frames in this sequence are not available (exceeds the time limits of the video), a uniform image with a temperature of 22°C was used instead. Each of the three consequent patches in this set was combined into a single RGB patch and was fed to the backbone. This allows the use of pre-trained backbone weights as well as reduced run-time in comparison to the option of feeding each patch separately to the backbone. Similarly to *Carion et al., 2020*, each of the backbone's output feature vectors was attached with a positional encoder. However, we adjusted the positional encoding to include additional information on the time of each feature vector (in addition to its spatial location). To do that, we computed time encoding in the same way it was computed by *Carion et al., 2020* for encoding the *x* or *y* coordinate and concatenated it to the *x, y* position encoding vector. To keep the length of the joint position and time encoding the same, we added a fully connected trainable layer that gets the (*x, y, t*) embedding as input (dim = 128*3 = 384) and outputs a feature vector with dim = 256 which allows using the rest of the neural network and pre-trained weights without additional changes. Lastly, instead of using 100 queries as in *Carion et al., 2020*, we used just a single query to get just the classification of the input set of patches and disabled the computation of a bounding box. Since our training set is relatively small, we used transfer learning and initialized the learnable weights with the weights published by *Carion et al., 2020* (weight file: detr-r50-dc5-f0fb7ef5.pth). We used the dc5 (dilated C5 stage) option proposed by *Carion et al., 2020*, which increases the spatial resolution of the backbone's output by a factor of 2 as it may be more suitable for classifying small objects, and used ResNet-50 as the backbone. We first trained the classifier using 39 train videos (each video contains a single experiment and includes both the habituation and trial periods and is of length of roughly 20 min). A second round of training used the weights of the first round as initial weights and included an additional 58 training videos (a total of 97 training videos).

Training database generation included extraction of (a) Positive examples of urine and feces that were manually marked. (b) Forty negative examples (labeled as background) per video in randomly selected positions and time (half during habituation and half during trial) that are not close in space and time to any manual annotation. (c) Hard negative examples consist of preliminary detected blobs (detected by the heuristic detection algorithm) that are not close in space and time to any manual detection. For both types of negative examples, a negative example in position $x_d$ and time $t_d$ was considered to be close to a manual detection of position $x_m$ in time $t_m$ if $distance(x_d, x_m) < 25 pixels$ and $-10sec \leq t_d - t_m \leq 30sec$. For the positive examples, we augmented the data by a time shift of [−3s..6s], compensating for possible differences between the manual tagging and the preliminary detection time, as well as increasing the training set size. Data augmentation for all examples included a random spatial shift of +−2 pixels, random flip, and rotation of 90°, 180°, and 270°. Input data was normalized to contain values between [0..255] using linear mapping that mapped 10°C to 0 and 40°C to 255. Values that exceeded 0 or 255 were trimmed. The first training round (39 training videos) was done for 230 epochs with a learning rate of 1e−5 for the backbone and 1e−4 for the rest of the weights and a factor 10 learning rate drop after 200 epochs. The second training round (97 training videos) was done for 50 epochs with a learning rate of 1e−5 for the backbone and 1e−4 for the rest of the weights and a factor 10 learning rate drop after 40 epochs.

## Accuracy measurement

The accuracy of automatic detections was evaluated using the following principles: (1) Manually tagged urine or fecal deposition is considered correctly detected by the algorithm, if an automatic detection with the same label exists at a distance of up to 20 pixels (2.9 cm) and in a time difference of up to 15 s. Spatial tolerance is required due to inherent ambiguity in the manual urine tagging process, as different observers often mark large spots or long traces of urine differently (see *Figure 2d* for an example of such a trace). Specifically, the detection algorithm might unify adjacent urine spots, tagged as multiple urine depositions by human annotators (see e.g. *Figure 2—video 1* and *Figure 3—figure supplement 3*). Temporal tolerance is required as the mouse body may cover the deposit or be very close to it for a while, thus delaying the time the preliminary detection algorithm detects it. (2) In the case described in 1, all automatic detections in this time and space window that got a correct label by the algorithm as the manual tagging are not counted as false alarms. (3) In contrast, if only automatic detections carrying labels different from the manually tagged deposition exist in the relevant space and time around it, then the closest one will be associated with this manual annotation and will be counted as misclassification (i.e., urine that was classified as feces or BG and feces that was classified as urine or BG), while the others will be counted as false alarms (will be counted in the BG column of the confusion matrix).

## Comparison with a second human annotator

The task of detecting and correctly classifying urine and feces in thermal videos is also challenging for a human annotator. To assess the performance of the DeePosit algorithm and compare it to a human annotator, 25 test videos were manually annotated by a second human annotator that marked a polygon surrounding each feces or urine spot. The detections of the DeePosit algorithm and of the second human annotator were compared to the annotation of the first human annotator (see *Figure 3f, g*).

## Comparison with YOLOv8 object detector

We compared our algorithm with a YOLOv8 (*Jocher et al., 2023*) based algorithm (YOLOv8n architecture). We trained YOLOv8 on 39 thermal video clips that were manually tagged with bounding boxes around each feces or urine spot. An additional 25 videos were annotated with bounding boxes for validation. OpenLabeling annotation tool was used for bounding boxes annotation (*Cartucho et al., 2018*). The training was done for 10,000 epochs with default parameters. Weights were initialized with YOLOv8n.pt pre-trained weight file, which was published by *Jocher et al., 2023*. Output weight file with the best accuracy on the validation videos was chosen. As YOLOv8 expects the pixel values to be between 0 and 255, temperatures between 10 and 40°C were linearly mapped to values between 0 and 255. As YOLOv8 is designed for 3-channel RGB images, we compared two training approaches. The first approach (termed YOLOv8 Gray) used the same thermal image for the R, G, and B channels. The second approach used three thermal images from time $t$, $t + 10$ s, and $t + 30$ s, where $t$ is the time of the deposition tagging, and fed them to the YOLOv8 classifier as the R, G, and B channels. This gives the classifier relevant temporal information that might capture the cool-down process, smearing of urine or shift of feces. Training examples included all frames in which a manual detection was labeled. Bounding boxes were annotated around all warm and clearly visible urine or feces in each of these frames (including old urine and feces that are still warm and clearly visible). In addition, 40 randomly selected images (from each training video) with no manual detection in a time period of −60…+10 s were added to the training set. During inference, YOLOv8 Gray or YOLOv8 RGB was activated on each frame of the thermal video. To prevent the same deposition from being detected many times, overlapping detections with the same label were unified if no more than 30 s passed between them. We compared the accuracy achieved by YOLOv8 Gray and YOLOv8 RGB with the DeePosit algorithm that was trained on the same 39 training videos. The results are shown in *Figure 3h–j*.

## Model evaluation on mice of a different strain (C57BL/6)

To evaluate the usability of our method in a different strain of mice and a different setting, we conducted 10 SP and 10 SxP experiments with C57BL/6 black mice using a white Plexiglass box arena (37 cm × 22 cm × 35 cm). We used the same classifier and the same preliminary detection parameters.

Note that the training set does not include C57BL/6 mice videos or videos with white arenas (see *Figure 3—figure supplement 1*, *Figure 4—figure supplement 1*, *Figure 5d, e*, and *Figure 5—figure supplement 1d, e* for results).

## Statistical analysis

We used a two-sided Wilcoxon rank sum test (Matlab's *ranksum* function) for all pairwise comparisons. Rank sum p-value equal to or smaller than 0.1, 0.05, 0.01, and 0.001 was marked with #, *, **, and ***, respectively. In addition, since some of the data is zero-inflated (many mice do not deposit urine or feces in the relevant measured period), we used a two-way Chi-square test to compare the distribution of zeros and non-zeros in the male group versus the female group in *Figure 6* and in *Figure 6—figure supplement 1*. The two-way Chi-square test was implemented using Matlab (see code in Appendix 1). p-value equal or smaller than 0.1, 0.05, 0.01, and 0.001 was marked with !, +, ++, and +++, respectively, and was mentioned to the left side of the ranksum p-value symbol (i.e, the notation +/** means that two-way Chi-square test resulted in p-value ≤0.05 and the ranksum test resulted in p-value ≤0.01). For the habituation versus trial comparison (*Figure 5a, b*, *Figure 5—figure supplement 2*), and the side preference analysis (*Figure 4—figure supplement 2*), mice with zero urine detections across all periods of the same test were ignored. The same was done for the feces analysis. Lastly, we used Matlab's *kruskalwallis* function for the Krusukal–Wallis test, which was used to examine the effect of test type (SP, SxP, and ESPs) on the dynamics of the urine and feces rate (*Table 1*) and area (*Appendix 1—table 1*). Additional statistical data for the figures is available. No mouse selection was done, but several videos in which the arena was not positioned well below the camera were excluded. Two cohorts of CD1 mice were used, each of which included at least eight mice and was tested by a different experimenter.

## Acknowledgements

We thank Prof Lucy Osborne and Prof Giuseppe Testa for providing the *Gtf2i+/Dup* mutant mice line. We also thank Yaniv Goldstein, Janet Tabakova, Wjdan Awaisy, and Shorook Amara for their help in annotating the videos and Sara Sheikh for drawing the experiment setup illustration. This study was supported by ISF-NSFC joint research program (Grant No. 3459/20), the Israel Science Foundation (Grants No. 1361/17 and 2220/22), the Ministry of Science, Technology and Space of Israel (Grant No. 3-12068), the Ministry of Health of Israel (Grant #3-18380 for EPINEURODEVO), the German Research Foundation (DFG) (GR 3619/16-1 and SH 752/2-1), the Congressionally Directed Medical Research Programs (CDMRP) (Grant No. AR210005) and the United States-Israel Binational Science Foundation (Grant No. 2019186).

## Additional information

### Funding

| Funder | Grant reference number | Author |
| --- | --- | --- |
| ISF-NSFC joint research program | 3459/20 | Shlomo Wagner |
| Israel Science Foundation | 1361/17 | Shlomo Wagner |
| Israel Science Foundation | 2220/22 | Shlomo Wagner |
| Ministry of Science, Technology and Space of Israel | 3-12068 | Shlomo Wagner |
| Ministry of Health of Israel | 3-18380 | Shlomo Wagner |
| German Research Foundation | GR 3619/16-1 | Shlomo Wagner |
| German Research Foundation | SH 752/2-1 | Shlomo Wagner |

| Funder | Grant reference number | Author |
| --- | --- | --- |
| Congressionally Directed Medical Research Programs | AR210005 | Shlomo Wagner |
| United States-Israel Binational Science Foundation | 2019186 | Shlomo Wagner |

The funders had no role in study design, data collection, and interpretation, or the decision to submit the work for publication.

## Author contributions

David Peles, Formal analysis, Writing – original draft, Algorithm development, Video annotation; Shai Netser, Writing – review and editing, Algorithm development, Experimental setup building; Natalie Ray, Data curation, Performing social behaviour tests, Video annotation; Taghreed Suliman, Data curation, Performing social behaviour tests, Video annotation; Shlomo Wagner, Resources, Funding acquisition, Writing – review and editing

## Author ORCIDs

David Peles (ID) https://orcid.org/0009-0003-5543-1676
Shai Netser (ID) https://orcid.org/0000-0003-4176-1124
Taghreed Suliman (ID) https://orcid.org/0009-0009-4177-5391
Shlomo Wagner (ID) https://orcid.org/0000-0002-7618-0752

## Ethics

All experiments were approved by the University of Haifa ethics committee (Reference #: UoH-IL-2301-103-4).

Reviewer #1 (Public review): https://doi.org/10.7554/eLife.100739.3.sa1
Author response https://doi.org/10.7554/eLife.100739.3.sa2

---

# Additional files

## Supplementary files

MDAR checklist

## Data availability

Raw video files will be shared with any academic PI upon a reasonable request from the corresponding author (davidpelz@gmail.com) and within a reasonable time. The data is free to use for academic purposes. No project proposal is required. For commercial use, please contact Shlomo Wagner ( shlomow@research.haifa.ac.il). The code, trained classifier weights, statistical data, and an example of a raw video are shared in GitHub (copy archived at *Peles, 2025*) and Zenodo.

The following dataset was generated:

| Author(s) | Year | Dataset title | Dataset URL | Database and Identifier |
| --- | --- | --- | --- | --- |
| David P, Shai S, Natalie R, Taghreed S, Shlomo W | 2025 | DeePosit: an AI-based tool for detecting mouse urine and fecal depositions from thermal video clips of behavioral experiments | https://doi.org/ 10.5281/zenodo. 14754159 | Zenodo, 10.5281/ zenodo.14754159 |

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

## Appendix 1

Code for computing Two Way Chi-Square Test which was used to compare the distribution of active mice (with at least one detection) in males vs females.

```
% Compute two way chi square test for 2x2 table
%
% Hypothseis H0: there is no relation between gender and the distribution
of zeros .
% Hypothseis H1: there is a relation between gender and the distribution of
zeros .
%
% Inputs :
% valsMales - a vector of length 2 that contains : [zeros count , non zeros
count] for males .
% valsFemales - a vector of length 2 that contains : [zeros count , non
zeros count] for females.
%
% Outputs :
% pVal - p value . A value lower than 0.05 suggests that the hypothesis H0
should be rejected .
% chiStat - statistic of the chi square test .
% df - degree of freedom (equals 1 for 2x2 tables).
%
function [pVal , chiStat ,df] = TwoWayChiSqrTest (valsMales , valsFemales)
if length (valsMales)~=2 || length (valsFemales)~=2
⎡error ('input vectors should have length =2 ')
end
sumMales = sum (valsMales);
sumFemales = sum (valsFemales);
sumAll = sumMales + sumFemales ;
sum1 = valsMales (1) + valsFemales (1) ;
sum2 = valsMales (2) + valsFemales (2) ;
expectedFreqMales = [sumMales *(sum1 / sumAll), sumMales *(sum2 / sumAll)];
expectedFreqFemales = [sumFemales *(sum1 / sumAll), sumFemales *(sum2 /
sumAll)];
chiStatMales = sum ((valsMales - expectedFreqMales).^2 ./
expectedFreqMales);
chiStatFemales = sum ((valsFemales - expectedFreqFemales).^2 ./
expectedFreqFemales);
chiStat = chiStatMales + chiStatFemales ;
df = 1;
pVal = 1- chi2cdf (chiStat ,df);
```

**Appendix 1—table 1.** The effect of the test on the urine and feces area.
Kruskal–Wallis test was used to check if the test type (SP, SxP, and ESPs) affects the area of urine or feces. p-value equal to or smaller than 0.1, 0.01, 0.001 was marked with #, **, ***, respectively. For male mice, $n = 24, 28, 21$ for SP, SxP, and ESPs. For female mice, $n = 15, 16, 17$ accordingly.

| Measurement | Habituation, minutes 1–4 | Habituation, minutes 11–14 | Trial, minute 1 | Trial, minutes 2–4 |
|---|---|---|---|---|
| Male urine area | 0.0003*** | 0.3436 | 0.0011** | 0.0614# |
| Female urine area | 0.3847 | 0.374 | 0.399 | 0.3124 |

*Appendix 1—table 1 Continued on next page*

*Appendix 1—table 1 Continued*

| Measurement | Habituation, minutes 1–4 | Habituation, minutes 11–14 | Trial, minute 1 | Trial, minutes 2–4 |
| --- | --- | --- | --- | --- |
| Male feces area | 0.0098** | 0.3315 | 0.2738 | 0.8938 |
| Female feces area | 0.2352 | 0.5138 | 0.1553 | 0.571 |

