## [Editor Report · eLife Assessment]

This manuscript presents an **important** machine-learning-based approach to the automated detection of urine and fecal deposits by rodents, key ethological behaviors that have traditionally been very poorly studied. The strength of evidence for the claim is **solid**, showing accuracy near 90% across several contexts. Training and testing for the specific contexts used by other experimenters, however, is probably warranted to make the model most relevant to the data that may be analyzed.

---

## [Referee Report · Reviewer #1 (Public review)]

Summary:

The manuscript provides a novel method for the automated detection of scent marks from urine and feces in rodents. Given the importance of scent communication in these animals and their role as model organisms, this is a welcome tool.

Strengths:

The method uses a single video stream to allow for the distinction between urine and feces. It is automated.

Weaknesses:

The accuracy is decent but not perfect and may be too low to detect some effects that are biologically real but subtle (e.g. less than 10% differences). For many assays, however, this tools will be useful.

---

## [Author Response]

The following is the authors’ response to the original reviews

We thank the reviewers for their constructive and helpful comments, which led us to make major changes in the model and manuscript, including adding the results of new experiments and analyses. We believe that the revised manuscript is much better than the previous version and that it addresses all issued raised by the reviewers.

Summary of changes made in the revised manuscript:

(1) We increased the training set size from 39 video clips to 97 video clips and the testing set size from 25 video clips to 60 video clips. The increase in training set size improved the overall accuracy from a mean F1 score of 0.81 in the previous version to a mean F1 score of 0.891 (see Figure 2 and Figure 3) in the current version. Specifically, the F1 score for urine detection was improved from 0.79 to 0.88.

(2) We further evaluated the accuracy of the DeePosit algorithm in comparison to a second human annotator and found that the algorithm accuracy is comparable to human-level accuracy.

(3) The additional test videos allowed us to test the consistency of the algorithm performance across gender, space, time, and experiment type (SP, SxP, and ESPs). We found consistent levels of performance across all categories (see Figure 3), suggesting that errors made by the algorithm are uniform across conditions, hence should not create any bias of the results.

(4) In addition, we tested the algorithm performance on a second strain of mice (male C57BL/6) in a different environmental condition (white arena instead of a black one) and found that the algorithm achieves comparable accuracy, even though C57BL/6 mice and white arena were not included in the training set. Thus, the algorithm seems to be robust and efficient across various experimental conditions.

(5) Analyzing urination and defecation dynamics in an additional strain of mice revealed interesting strain-specific features, as discussed in the revised manuscript.

(6) Overall, we found DeePosit accuracy to be stable with no significant bias across stages of the experiment, types of the experiment, gender of the mice, strain of mice, and across experimental conditions.

(7) We also compared the performance of DeePosit to a classic object detection algorithm: YOLOv8. We trained YOLOv8 both on a single image input (YOLOv8 Gray) and on 3 image inputs representing a sequence of three time points around the ground truth event (t): t+0, t+10, and t+30 seconds (YOLOv8 RGB). DeePosit achieved significantly better accuracy over both YOLOv8 alternatives. YOLOv8 RGB achieved better accuracy than YOLOv8 Gray, suggesting that temporal information is important for this task. It's worth mentioning that while YOLOv8 requires the annotator to draw rectangles surrounding each urine spot or feces as part of the training set, our algorithm training set used just a single click inside each spot, allowing faster generation of training sets.

(8) As for the algorithm parameters, we tested the effect of the main parameter of the preliminary detection (the temperature threshold for the detection of a new blob) and found that a threshold of 1.6°C gave the best accuracy and used this parameter for all of the experiments instead of 1.1°C which was used in the original manuscript. It's worth mentioning that the performance is quite stable (mean F1 score of 0.88-0.89) for the thresholds between 1.1°C and 3°C (Figure 3—Figure Supplement 2).

(9) We also checked if changing the input length of the video clip that is fed to the classifier affects the accuracy by training the classifier with -11.30 seconds video clips (41 seconds in total) instead of -11.60 seconds (71 seconds in total) and found no difference in accuracy.

(10) In the revised paper, we report recall, precision, and F1 scores in the caption of the relevant figures and also supply Excel files with the full statistics for each of the figures.

**Public Reviews:**

**Reviewer #1 (Public Review):**
Summary:The manuscript provides a novel method for the automated detection of scent marks from urine and feces in rodents. Given the importance of scent communication in these animals and their role as model organisms, this is a welcome tool.

We thank the reviewer for the positive assessment of our tool

Strengths:The method uses a single video stream (thermal video) to allow for the distinction between urine and feces. It is automated.Weaknesses:The accuracy level shown is lower than may be practically useful for many studies. The accuracy of urine is 80%.

We have trained the model better, using a larger number of video clips. The increase in training set size improved the overall accuracy from a mean F1 score of 0.81 in the previous version to a mean F1 score of 0.891 (see Figure 2 and Figure 3) in the current version. Specifically, the F1 score for urine detection was improved from 0.79 to 0.88.

This is understandable given the variability of urine in its deposition, but makes it challenging to know if the data is accurate. If the same kinds of mistakes are maintained across many conditions it may be reasonable to use the software (i.e., if everyone is under/over counted to the same extent). Differences in deposition on the scale of 20% would be challenging to be confident in with the current method, though differences of the magnitude may be of biological interest. Understanding how well the data maintain the same relative ranking of individuals across various timing and spatial deposition metrics may help provide further evidence for the utility of the method.

The additional test videos allowed us to test the consistency of the algorithm performance across gender, space, time and experiment type (SP, SxP, and ESP). We found consistent levels of performance across all categories (see Figure 3), suggesting that errors made by the algorithm are uniform across conditions, hence should not create any bias of the results.

**Reviewer #2 (Public Review):**
Summary:The authors built a tool to extract the timing and location of mouse urine and fecal deposits in their laboratory set up. They indicate that they are happy with the results they achieved in this effort.

Yes, we are.

The authors note urine is thought to be an important piece of an animal's behavioral repertoire and communication toolkit so methods that make studying these dynamics easier would be impactful.

We thank the reviewer for the positive assessment of our work.

Strengths:With the proposed method, the authors are able to detect 79% of the urine that is present and 84% of the feces that is present in a mostly automated way.Weaknesses:The method proposed has a large number of design choices across two detection steps that aren't investigated. I.e. do other design choices make the performance better, worse, or the same?

We chose to use a heuristic preliminary detection algorithm for the detection of warm blobs, since warm blobs can be robustly detected with heuristic algorithms without the need for a training set. This design selection might allow easier adaptation of our algorithm for different types of arenas. Another advantage of using a heuristic preliminary detection is the easy control of the preliminary detection parameters such as the minimum temperature difference for detecting a blob, size limits of the detected blob, cooldown rate and so on that may help in adopting it to new conditions. As for the classifier, we chose to feed it with a relatively small window surrounding each preliminary detection, and hence it is not affected by the arena’s appearance outside of its region of interest. This should allow lower sensitivity to the arena’s appearance.

As for the algorithm parameters, we tested the effect of the main parameter of the preliminary detection (the temperature threshold for the detection of a new blob) and found that a threshold of 1.6°C gave the best accuracy and used this parameter for all of the experiments instead of 1.1°C which was used in the original manuscript. It's worth mentioning that the performance is quite stable (mean F1 score of 0.88-0.89) for the thresholds between 1.1°C and 3°.

We also checked if changing the input length of the video clip fed to the classifier affects the accuracy by training the classifier with -11.30 seconds video clips (41 seconds in total) instead of -11.60 seconds (71 seconds in total) and found no difference in accuracy.

Overall, the algorithm's accuracy seems to be rather stable across various choices of parameters.

Are these choices robust across a range of laboratory environments?

We tested the algorithm performance on a second strain of mice (male C57BL/6) in a different environmental condition (white arena instead of a black one) and found that the algorithm achieves comparable accuracy, even though C57BL/6 mice and white arena were not included in the training set. Thus, the algorithm seems to be robust and efficient across various experimental conditions.

How much better are the demonstrated results compared to a simple object detection pipeline (i.e. FasterRCNN or YOLO on the raw heat images)?

We compared the performance of DeePosit to a classic object detection algorithm: YOLOv8. We trained YOLOv8 both on a single image input (YOLOv8 Gray) and on 3 image inputs representing a sequence of three time points around the ground truth event (t): t+0, t+10, and t+30 seconds (YOLOv8 RGB). DeePosit achieved significantly better accuracy over both YOLOv8 alternatives. YOLOv8 RGB achieved better accuracy than YOLOv8 Gray, suggesting that temporal information is important for this task. It's worth mentioning that while YOLOv8 requires annotator to draw rectangles surrounding each urine spot or feces as part of the training set, our algorithm training set used just a single click inside each spot, allowing faster generation of a training sets.

The method is implemented with a mix of MATLAB and Python.

That is right.

One proposed reason why this method is better than a human annotator is that it "is not biased." While they may mean it isn't influenced by what the researcher wants to see, the model they present is still statistically biased since each object class has a different recall score. This wasn't investigated. In general, there was little discussion of the quality of the model.

We tested the consistency of the algorithm performance across gender, space, time and experiment type (SP, SxP, and ESP). We found consistent levels of performance across all categories (see Figure 3), suggesting that errors made by the algorithm are uniform across conditions, hence should ne create any bias of the results. Specifically, the detection accuracy is similar between urine and feces, hence should not impose a bias between the various object classes.

Precision scores were not reported.

In the revised paper we report recall, precision, and F1 scores in the caption of the relevant figures and also supply Excel files with the full statistics for each of the figures.

Is a recall value of 78.6% good for the types of studies they and others want to carry out? What are the implications of using the resulting data in a study?

We have trained the model better, using a larger number of video clips. The increase in training set size improved the overall accuracy from a mean F1 score of 0.81 in the previous version to a mean F1 score of 0.891 (see Figure 2 and Figure 3) in the current version. Specifically, the F1 score for urine detection was improved from 0.79 to 0.88.

How do these results compare to the data that would be generated by a "biased human?"

We further evaluated the accuracy of the DeePosit algorithm in comparison to a second human annotator and found that the algorithm accuracy is comparable to human-level accuracy (Figure 3).

5 out of the 6 figures in the paper relate not to the method but to results from a study whose data was generated from the method. This makes a paper, which, based on the title, is about the method, much longer and more complicated than if it focused on the method.

We appreciate the reviewer's comment, but the analysis of this new dataset by DeePosit demonstrates how the algorithm may be used to reveal novel and distinguishable dynamics of urination and defecation activities during social interactions, which were not yet reported.

Also, even in the context of the experiments, there is no discussion of the implications of analyzing data that was generated from a method with precision and recall values of only 7080%. Surely this noise has an effect on how to correctly calculate p-values etc. Instead, the authors seem to proceed like the generated data is simply correct.

As mentioned above, the increase in training set size improved the overall accuracy from a mean F1 score of 0.81 in the previous version to a mean F1 score of 0.891 (see Figure 2 and Figure 3) in the current version. Specifically, the F1 score for urine detection was improved from 0.79 to 0.88.

**Reviewer #3 (Public Review):**
Summary:The authors introduce a tool that employs thermal cameras to automatically detect urine and feces deposits in rodents. The detection process involves a heuristic to identify potential thermal regions of interest, followed by a transformer network-based classifier to differentiate between urine, feces, and background noise. The tool's effectiveness is demonstrated through experiments analyzing social preference, stress response, and temporal dynamics of deposits, revealing differences between male and female mice.Strengths:The method effectively automates the identification of depositsThe application of the tool in various behavioral tests demonstrates its robustness and versatility.The results highlight notable differences in behavior between male and female mice

We thank the reviewer for the positive assessment of our work.

Weaknesses:The definition of 'start' and 'end' periods for statistical analysis is arbitrary. A robustness check with varying time windows would strengthen the conclusions.

In all the statistical tests conducted in the revised manuscript, we have used a time period of 4 minutes for the analysis. We did not used the last minute of each stage for the analysis since the input of DeePosit requires 1 minute of video after the event. Nevertheless, we also conducted the same tests using a 5-minute period and found similar results (Figure 5—Figure Supplement 1).

The paper could better address the generalizability of the tool to different experimental setups, environments, and potentially other species.

As mentioned above, we tested the algorithm performance on a second strain of mice (male C57BL/6) in a different environmental condition (white arena instead of a black one) and found that the algorithm achieves comparable accuracy, even though C57BL/6 mice and white arena were not included in the training set. Thus, the algorithm seems to be robust and efficient across various experimental conditions.

The results are based on tests of individual animals, and there is no discussion of how this method could be generalized to experiments tracking multiple animals simultaneously in the same arena (e.g., pair or collective behavior tests, where multiple animals may deposit urine or feces).

At the moment, the algorithm cannot be applied for multiple animals freely moving in the same arena. However, in the revised manuscript we explicitly discussed what is needed for adapting the algorithm to perform such analyses.

**Recommendations for the authors:**
- Add a note and/or perform additional calculations to show that the results do not depend on the specific definitions of 'start' and 'end' periods. For instance, vary the time window thresholds and recalculate the statistics using different windows (e.g., 1-5 minutes instead of 1-4 minutes).

In all the statistical tests conducted in the revised manuscript, we have used a time period of 4 minutes for the analysis. We did not use the last minute of each stage for the analysis since the input of DeePosit requires 1 minute of video after the event. Nevertheless, we also conducted the same tests using a 5-minute period and found similar results (Figure 5—Figure Supplement 1).

- Condense Figures 4, 5, and 6 to simplify the presentation. Focus on demonstrating the effectiveness of the tool rather than detailed experimental outcomes, as the primary contribution of this paper is methodological.

We have added to the revised manuscript one technical figure (Figure 3) comparing the accuracy of the algorithm performance across gender, space, time, and experiment type (SP, SxP, and ESP) as well as comparing its performance to a second human annotator and to YOLOv8. One more partially technical figure (Figure 5) compares the results of the algorithm between white ICR mice in the black arena and black C57BL/6 mice in the white arena. Thus, only Figures 4 and 6 show detailed experimental outcomes.

- Provide more detail on how the preliminary detection procedure and parameters might need adjustment for different experimental setups or conditions. Discuss potential adaptations for field settings or more complex environments.

As for the algorithm parameters, we tested the effect of the main parameter of the preliminary detection (the temperature threshold for the detection of a new blob) and found that a threshold of 1.6°C gave the best accuracy and used this parameter for all of the experiments instead of 1.1°C which was used in the original manuscript. It's worth mentioning that the performance is quite stable (mean F1 score of 0.88-0.89) for the thresholds between 1.1°C and 3°.

We also checked if changing the input length of the video clip that is fed to the classifier affects the accuracy by training the classifier with -11.30 seconds video clips (41 seconds in total) instead of -11.60 seconds (71 seconds in total) and found no difference in accuracy.

Overall, the algorithm's accuracy seems to be rather stable across various choices of parameters.

Editor's note:Should you choose to revise your manuscript, please ensure your manuscript includes full statistical reporting including exact p-values wherever possible alongside the summary statistics (test statistic and df) and 95% confidence intervals. These should be reported for all key questions and not only when the p-value is less than 0.05 in the main manuscript.

We have deposited the detailed statistics of each figure in https://github.com/davidpl2/DeePosit/tree/main/FigStat/PostRevision